# Multiparameter immunoprofiling for the diagnosis and differentiation of progressive versus nonprogressive nontuberculous mycobacterial lung disease–A pilot study

Paige K. Marty[1]ᵒ, Balaji Pathakumari[1]ᵒ, Thomas M. Cox[1], Virginia P. Van Keulen[1,2], Courtney L. Erskine[2], Maleeha Shah[1], Mounika Vadiyala[1], Pedro Arias-Sanchez[1], Snigdha Karnakoti[1], Kelly M. Pennington[1], Elitza S. Theel[3], Cecilia S. Lindestam Arlehamn[4], Tobias Peikert[1,2], Patricio Escalante[1] *

1 Division of Pulmonary and Critical Care Medicine, Department of Medicine, Mayo Clinic, Rochester, MN, United States of America, 2 Department of Immunology, Mayo Clinic, Rochester, MN, United States of America, 3 Department of Laboratory Medicine, Mayo Clinic, Rochester, MN, United States of America, 4 Center for Infectious Disease and Vaccine Research, La Jolla Institute for Immunology, La Jolla, CA, United States of America

ᵒ These authors contributed equally to this work.
* Escalante.Patricio@mayo.edu

## Abstract

Clinical prediction of nontuberculous mycobacteria lung disease (NTM-LD) progression remains challenging. We aimed to evaluate antigen-specific immunoprofiling utilizing flow cytometry (FC) of activation-induced markers (AIM) and IFN-γ enzyme-linked immune absorbent spot assay (ELISpot) accurately identifies patients with NTM-LD, and differentiate those with progressive from nonprogressive NTM-LD. A Prospective, single-center, and laboratory technician-blinded pilot study was conducted to evaluate the FC and ELISpot based immunoprofiling in patients with NTM-LD (n = 18) and controls (n = 22). Among 18 NTM-LD patients, 10 NTM-LD patients were classified into nonprogressive, and 8 as progressive NTM-LD based on clinical and radiological features. Peripheral blood mononuclear cells were collected from patients with NTM-LD and control subjects with negative QuantiFERON results. After stimulation with purified protein derivative (PPD), mycobacteria-specific peptide pools (MTB300, RD1-peptides), and control antigens, we performed IFN-γ ELISpot and FC AIM assays to access their diagnostic accuracies by receiver operating curve (ROC) analysis across study groups. Patients with NTM-LD had significantly higher percentage of CD4⁺/CD8⁺ T-cells co-expressing CD25⁺CD134⁺ in response to PPD stimulation, differentiating between NTM-LD and controls. Among patients with NTM-LD, there was a significant difference in CD25⁺CD134⁺ co-expression in MTB300-stimulated CD8⁺ T-cells (p <0.05; AUC-ROC = 0.831; Sensitivity = 75% [95% CI: 34.9–96.8]; Specificity = 90% [95% CI: 55.5–99.7]) between progressors and nonprogressors. Significant differences in the ratios of antigen-specific IFN-γ ELISpot responses were also seen for RD1-nil/PPD-nil and RD1-nil/anti-CD3-nil between patients with nonprogressive vs. progressive NTM-LD. Our results suggest that multiparameter immunoprofiling can accurately identify patients

**Data Availability Statement:** All relevant data and its Supporting Information files are within the manuscript.

**Funding:** This work was supported by the Lucile Nelson Career Development Award in Pulmonary Research. Part of this work was supported by the CHEST Foundation (K.M.P., P.E.), and the National Institute of Allergy and Infectious Diseases at the National Institutes of Health [AI141591 to P.E.]. Part of this project was also supported by Grant Number UL1 TR000135 from the National Center for Advancing Translational Sciences (NCATS). The funders had no role in study design, data collection and analysis, decision to publish, or preparation of the manuscript.

**Competing interests:** P.E. and T.P., and their institution have filed two patent applications related to immunodiagnostic laboratory methodologies for latent tuberculosis infection (Patent numbers: 9678071 and 10401360), which are not included in this manuscript. To date, there has been no income or royalties associated with those filed patent applications. This does not alter our adherence to PLOS ONE policies on sharing data and materials. PE participated in a short-term advisory scientific board for DiaSorin Molecular in 2020, which was outside the scope of the submitted manuscript, and honorarium was paid to Mayo Clinic. E.S.T serves as a consultant for Roche Diagnostics (Basel, Switzerland), Euroimmun US (Mountain Lakes, NJ, USA), and Seriummune Inc. (Goleta, CA, USA) on topics outside the scope of this manuscript. P.E., T.P., and E.S.T. have no other conflicts to declare. P.K.M., B.P., T.M.C., V.P.V, C. L.E., M.S., M.V., P.A.S., S.K., K.M.P., and C. S. L. A. have no conflicts to declare.

with NTM-LD and may identify patients at risk of disease progression. A larger longitudinal study is needed to further evaluate this novel immunoprofiling approach.

## Introduction

Nontuberculous mycobacteria (NTM) pathogens are causing more disease burden than tuberculosis (TB) in several regions of the world. The annual incidence and prevalence of NTM disease from 2008 to 2015 increased from 3.13 to 4.73 and 6.78 to 11.70 per 100 000/ year respectively in the United States [1]. NTM lung disease (NTM-LD) is the most common presentation accounting for 80–90% of all cases [2]. *Mycobacterium avium* complex (MAC) and *Mycobacterium abscessus* complex (MAbsC), are among the most common etiologies of NTM lung disease (NTM-LD) [3–5]. Many challenges are associated with managing patients with NTM-LD, including suboptimal diagnosis and non-guidelines-based treatments which are often difficult to tolerate. The clinical spectrum of NTM pulmonary infections is also wide, and the relationship between exposure, airway colonization, as well as nonprogressive and progressive infections with clinical and/or radiological disease manifestations, is not well-defined. Symptoms associated with NTM-LD are often non-specific, and the presence of NTM in a single sputum sample does not necessarily equate to active lung disease. Therefore, clinical and radiological findings, as well as microbiological criteria, are critical for the accurate diagnosis of NTM-LD [6–8]. Moreover, some patients are unable to provide adequate sputum samples, which can also preclude the timely diagnosis of NTM-LD. Importantly, many patients have limited disease severity without clinical and/ or overt radiological signs of progression over time. These patients may not require antimicrobial therapy and are often managed with active airway clearance treatment and close clinical-radiological surveillance [9]. When the infection progresses, NTM causes inflammation of the airways and lung tissue which can lead to progressive lung damage and systemic illness.

Currently, there are no diagnostic tests to *a priori* differentiate progressive from nonprogressive NTM-LD, and the risk factors of disease progression remain unclear. In fact, the clinical assessment of these patients is frequently complicated by the presence of other comorbidities, including pulmonary diseases such as chronic obstructive pulmonary disease (COPD) and bronchiectasis [10]. Therefore, the development of non-sputum-based biomarkers and methods that accurately diagnose NTM-LD and reliably identify patients at risk of progressive disease are urgently needed. Such new blood-based NTM diagnostics would facilitate risk stratification and inform the timely individualized management of patients with nonprogressive and progressive forms of NTM-LD [11].

Flow cytometric (FC) detection of activation-induced markers (AIM) in T-cells after *ex vivo* antigen challenge in peripheral blood mononuclear cells (PBMC) can differentiate the state of infection due to *M. tuberculosis* [12–15]. We previously reported that combinatory immunoprofiling using FC detection of AIM CD25$^+$ and CD134$^+$ in T-cells following *ex vivo* antigen challenge in PBMCs can differentiate treated and untreated latent TB infection and might identify patients at risk for future TB reactivation [12]. Herein we conducted a pilot study to test whether a similar immunoprofiling approach can accurately identify and differentiate patients with nonprogressive and progressive NTM-LD, which can potentially assist to individualize treatment management for these patients.

## Materials and methods

### Study participants

The study was approved by the Mayo Clinic Institutional Review Board (IRB)(IRB#:09–003253). All study participants signed a written informed consent and were enrolled in Mayo Clinic, Rochester, Minnesota between August 2017 and November 2021. All the methods were carried out in accordance with relevant guidelines and regulations after obtaining approval and recommendations from the Mayo Clinic IRB. This pilot study was part of a prospective, single-center, laboratory technician-blinded larger study to investigate new immune biomarkers in tuberculosis, which includes patients with NTM-LD as control subjects. We recruited NTM-LD patients, including patients with clinical and radiological signs of progressive diseases as well as patients with nonprogressive NTM-LD. We obtained the patient clinical data from their medical records: age, sex, ethnicity, smoking history, usage of immunosuppressive drugs, history of TB and other underlying pulmonary diseases, co-morbidities, radiographical features, BMI, bacterial culture results, pulmonary function and other laboratory test results. We applied the American Thoracic Society and the Infectious Diseases Society of America (ATS/IDSA) guidelines diagnostic criteria for NTM-LD [7]. We also included local subjects with negative QuantiFERON In-tube™ (QFT) results and no clinical evidence or radiological signs of NTM-PD as study controls. Nonprogressive NTM-LD was defined as clinical and radiological stability over at least 24 months in patients not treated with antibiotics for NTM-LD. Patients that were determined to have progressive NTM-LD had signs of clinical and/or radiological progression over at least 6 months and/or were judged by their clinician to receive antibiotic treatment for NTM-LD. Subjects under 18 years of age and individuals with latent TB infection (LTBI) were excluded from the study. Other study exclusion criteria included: patients living with HIV, active cancer, autoimmune diseases, or a history of solid organ or hematological transplantation.

### PBMC isolation

Peripheral blood from the enrolled study participants was collected in sodium-heparin tubes. PBMCs were isolated using the Ficoll density gradient separation protocol as previously described [12]. The viable cells were counted by hemocytometer and cryopreserved in the freezing media (10% DMSO with 90% cosmic calf serum) (Fisher Scientific, Hampton, NH).

### PBMCs *ex vivo* stimulation, and interferon-gamma (IFN-γ) ELISpot assay

To ensure reproducibility, all the samples for each stimulation were tested in triplicates. In a 96-well plate, $2.5 \times 10^5$ cells/well were stimulated with ESAT-6 (4 μg/mL)/CFP-10 (2 μg/mL) (region of deletion or RD1) peptides, purified protein derivative (PPD-10 μg/mL) (AJ vaccines, Denmark), MTB300 peptide pool (0.5 μg/mL) (gift from Dr. Cecellia L. Arlehamn, La Jolla Institute for Immunology, LJ, CA), Candida antigen (2 μg/mL) (Mybiosource, San Diego, CA), anti-CD3 antibody (100 ng/mL) (ThermoFisher Scientific, Waltham, MA) and incubated at 37˚C for 24 hours. ELISpot plates (Millipore, Billerica, MA) were coated with 10 μg/mL IFN-γ capture antibody (clone 1-D1K; Mabtech, Mariemont, OH) and incubated overnight at 4˚C. After 24 hours, the ELISpot plates were washed with PBS and blocked with medium for 2 hours. The cells and supernatant were transferred from 96-well plates to the ELISpot plates and further incubated at 37˚C for 24 hours. After washing with PBS containing 0.05% tween-20 (PBST), 2 μg/mL of biotinylated secondary antibody for IFN-γ (clone 7-B6-1; Mabtech, Mariemont, OH) was added and the plates were incubated for 2 hours at 37˚C followed by another wash. Next, 1 μL/mL Streptavidin-horseradish peroxidase (BD Pharmingen, San

Diego, CA) in 10% FBS in phosphate-buffered saline (PBS) was added, and the plates were incubated for 1 hour at room temperature. For the final wash, plates were first washed with PBST, followed by washing with PBS. Plates were developed by adding AEC (3-amino-9-ethyl-carbazole) chromogen substrate (Sigma-Aldrich, St. Louis, MO) and the reaction was stopped with water. After drying overnight, the plates were read on an AID ELISpot reader (Autoimmun Diagnostika GmbH, Strassberg, Germany). ELISpot results were determined by measuring the mean spot forming units (sfu) frequency of the antigen-stimulated sample minus the mean sfu frequency of the unstimulated sample (nil).

## PBMCs *ex vivo* stimulation, immunostaining, and flow cytometry

The cryopreserved PBMCs were thawed and cultured in Rosewell Park Memorial Institute (RPMI) 1640 medium (Sigma-Aldrich, St. Louis, MO). Then the cells were stimulated with co-stimulatory antibodies (CD28, CD49d - 1μg/mL) (BD Bioscience, San Diego, CA) and test peptides/antigens such as ESAT-6/CFP-10 peptides, PPD, MTB300 peptide pool, Candida antigen, and anti-CD3 with specified concentrations as described above. Cells cultured under similar conditions without any stimulation served as a negative (nil) control. The culture plate was incubated for 40 hours at 37˚C with 5% $CO_2$, and the cells were stained with specific surface markers of anti-human monoclonal antibodies such as CD4 BV650 (BD Bioscience, San Diego, CA), and CD3 APC-AF750, CD8 krome Orange, CD25 APC-AF700, CD134 APC (Beckman Coulter, Brea, CA). Finally, cells were washed and fixed with 0.5% paraformaldehyde and at least 250,000 cells were acquired by BD LSRFortessa (BD Bioscience, San Diego, CA). Files were exported in FCS 3.0 format and the percentage of phenotypic markers was analyzed by Kaluza software (Beckman Coulter, Brea, CA). The background (nil) response was subtracted from the tested antigen stimulations. Considering the cell count, number of tested antigens, reagent cost and that this is a small pilot study, we performed the flow cytometry in a single well manner for each antigenic condition.

## Data analysis

Results were compared using the Chi-square test for categorical variables (Fisher exact test for cells with numbers ≤5). Data are expressed in the tables as numbers and percentages or medians with interquartile ranges (IQR), as appropriate. Non-parametric Kruskal–Wallis ANOVA with Dunn's post-test comparison was used to compare IFN-γ ELISpot, and FC antigen-specific results in controls, nonprogressive and progressive NTM-LD patients. The Mann-Whitney U-test was used to compare the data between the cohorts. Receiver operating characteristic (ROC) analysis was performed to define the diagnostic potential of markers by deriving the ROC area under curve (AUC) sensitivity and specificity at their 95% confidence intervals (CI) and best cut offs to differentiate the study groups. In stimulation experiments, percentage of activated T cells were adjusted by subtracting the unstimulated control value. The statistical difference was considered significant when the P values <0.05. We use Graph-Pad Prism 9.3.1 (GraphPad Software, San Diego, CA) for statistical analysis.

## Results

### Clinical characteristics of study participants

Ninety-two individuals were screened for study eligibility and 52 were excluded, including 44 subjects with LTBI (**Fig 1**). We enrolled 22 control subjects and 18 patients of NTM-LD, including 8 patients with progressive disease and 10 patients with nonprogressive NTM-LD that did not require antimicrobial therapy (**Tables 1, 2**). Controls subjects included 17 women

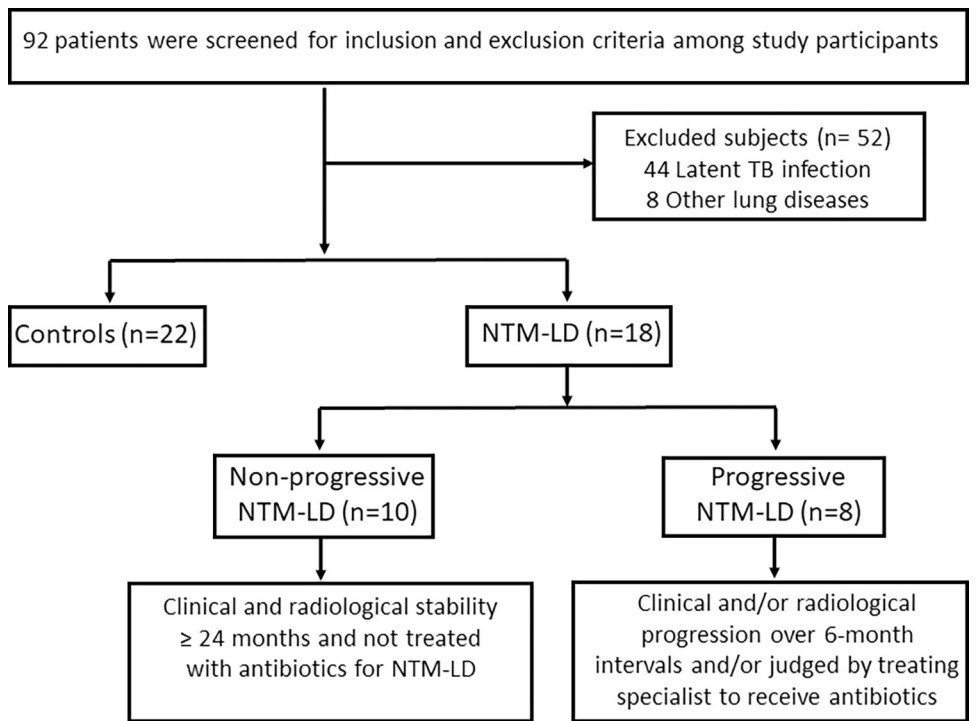

**Fig 1. Study flowchart of patient inclusion and exclusion criteria.**

and 5 men with no known history of *M. tuberculosis* infection and negative QFT results, along with no clinical or radiological signs of active pulmonary disease or NTM-LD. The demographics and baseline characteristics of the study groups were summarized in **Table 1**. There were significant differences in age between NTM-PD patients and controls, but no significant difference was found in sex, ethnicity, immunosuppression state, or smoking habits. There were no significant differences between nonprogressive and progressive NTM-LD in terms of weight loss, smoking, alcohol habits and co-morbidities. Similarly, we did not find significant differences between the groups in the laboratory assays like CRP, ESR, PFT, sputum and culture tests (**Table 2**). BMI was significantly lower in the progressive NTM-LD (19.1 kg/m2) compared with nonprogressive NTM-LD (25.5 kg/m2) (**Table 2**). *M. avium* was the most common species and nodular bronchiectasis was a frequent radiological feature in both the nonprogressive and progressive cohorts. Among 8 progressive NTM-LD, 6 patients had MAC and 2 had *M. abscessus* complex isolates. Of the 10 nonprogressive NTM-LD, 7 had MAC and others had either *M. abscessus* complex, *M. chelonae*, or *M. kansasii* species. BACES scores were similar across these two groups but four patients with progressive disease had cavitary lung lesions (**Table 2**).

## IFN-γ ELISpot for differentiating NTM-LD from control subjects

We compared the IFN-γ secreting T cells by ELISpot upon stimulation of PBMC with various antigens to differentiate the NTM infection status. The background (nil) IFN-γ secretion was subtracted from the stimulated PBMCs. The median levels of IFN-γ secreting T cells were significantly higher in NTM-LD patients than that in the control group (**Fig 2A-2D**). The diagnostic accuracy of ELISpot with these antigens are shown in **Table 3** and ROC analysis is depicted in **Fig 5A**. ELISpot with MTB300 stimulation showed the highest sensitivity of 78.5%

**Table 1. Demographic and clinical characteristics of study participants.**

| Demographic | Subjects, no. (%) | |
|---|---|---|
| | Controls (n = 22) | NTM-LD (n = 18) |
| **Age** | | |
| Median in years | 37 [30–65] | 66.5 [59–78] [a] |
| Range | 25–73 | 26–83 |
| **Sex** | | |
| Female | 17 (77.2) | 16 (88.9) |
| Male | 5 (22.7) | 2 (11.1) |
| **Ethnicity** | | |
| Caucasian | 20 (91) | 17 (94.4) |
| African American | 0 (0) | 0 (0) |
| Asian/Pacific Islander | 1 (4.5) | 1 (5.6) |
| Hispanic | 1 (4.5) | 0 (0) |
| Other | 0 (0) | 0 (0) |
| **Smoking history** | | |
| Never | 16 (72.7) | 10 (55.6) |
| Former | 6 (27.2) | 8 (44.4) |
| **Use of immunosuppressants** | | |
| No | 19 (86.3) | 15 (83.3) |
| Yes | 3 (13.6) | 3 (16.7) |
| **Healthcare worker** | | |
| No | 7 (31.8) | 7 (38.9) |
| Yes, no patient contact | 5 (22.7) | 0 (0) |
| Yes, patient contact or lab worker | 10 (45.4) | 11 (61.1) |
| **TB close contact** | | |
| Unlikely | 14 (63.6) | 13 (72.2) |
| Yes, remote (> 5 years) | 2 (9) | 1 (5.6) |
| Yes, recent (< 5 years) | 1 (4.5) | 0 (0) |
| Unknown | 5 (22.7) | 4 (22.2) |
| **Place of birth** | | |
| United States | 19 (86.3) | 16 (88.9) |
| Foreign born, TB endemic area | 1 (4.5) | 1 (5.6) |
| Foreign born, non-TB endemic area | 2 (9) | 1 (5.6) |
| **Cavitary lung disease** | | |
| No | N/A | 13 (72.2) |
| Yes | N/A | 4 (22.2) |

**Abbreviations:** TB = tuberculosis; NTM = non-tuberculous mycobacteria; IQR = Interquartile range; N/A = not applicable.

Values are number (%) or median [IQR].

[a] P <0.0013 for age comparison between groups by Mann-Whitney U test. Otherwise, no statistically significant differences between groups by Fisher's exact test or R x C exact test.

(95% CI 49.2–95.3) with 92.3% (95% CI, 63.9–99.8) specificity. ELISpot with the other RD1, PPD, and Candida antigens showed 64.2% (95% CI, 35.1–87.2) sensitivity and 80% (95% CI, 56.3–94.2) specificity. By subtracting the individual RD1 antigen ELISpot results from MTB300 (MTB300-RD1), we found a significantly higher IFN-γ sfu count for NTM-LD patients than controls (Fig 2E). In contrast, none of the ratios of ELISpot results with various antigen conditions of interest including RD1/PPD, MTB300/PPD, MTB300/Candida, PPD/

**Table 2. Clinical characteristics of patients with nonprogressive and progressive NTM-LD.**

| | Subjects, no. (%) | | |
|---|---|---|---|
| | **Non progressive NTM-LD (n = 10)** | **Progressive NTM-LD (n = 8)** | **P Value[a]** |
| **Demographics** | | | |
| **Female** | 10 (100) | 7 (87.5) | 0.44 |
| **Age in years** | 72.5 [61–79.8] | 61 [54–78.8] | 0.17 |
| **Ethnicity** | | | |
|   **White** | 10 (100) | 7 (87.5) | 0.44 |
|   **Asian** | 0 (0) | 1 (12.5) | 0.44 |
|   **Weight, kg** | 67.8 [56.4–75.1] | 53.8 [47.3–79.2] | 0.32 |
|   **BMI, kg/m2** | 25.5 [21.9–28.6] | 19.1 [17.8–22.6] | 0.05 |
| **Social History** | | | |
|   **Alcohol use** | 6 (60) | 4 (50) | 1 |
|   **Smoking history** | 6 (60) | 2 (25) | 0.19 |
| **Co-Morbidities** | | | |
|   **COPD** | 0 | 1 (12.5) | 0.44 |
|   **Asthma** | 0 | 1 (12.5) | 0.44 |
|   **Cardiovascular Disease** | 2 (20) | 3 (37.5) | 0.61 |
|   **Previous history of TB** | 0 | 0 | 1 |
|   **Diabetes Mellitus** | 1 (10) | 0 | 1 |
|   **Bronchiectasis** | 7 (70) | 6 (75) | 1 |
|   **Immunosuppression** | 1 (10) | 1 (12.5) | 1 |
|   **Malignancy** | 1 (10) | 0 (0) | 1 |
| **Pulmonary Function Test** | 4 (40) | 4 (50) | - |
|   **FEV1** | 62 [59.5–66.8] | 80 [33.3–100.5] | 0.68 |
|   **FVC** | 77 [70–100.5] | 88 [64.8–108.3] | 0.89 |
|   **FEV1/FVC ratio** | 69.25 [63.3–83.3] | 85 [75.5–101] | 0.23 |
|   **DLCO** | 74.5 [69.8–92] | 59.5 [30–92] | 0.34 |
|   **TLC** | 96 [82.3–120.3] | 100.5 [82.5–113.25] | 0.87 |
| **Symptoms** | | | |
|   **Cough** | 8 (80) | 7 (87.5) | 1 |
|   **Dyspnea** | 2 (20) | 1 (12.5) | 1 |
|   **Hemoptysis** | 0 (0) | 1 (12.5) | 0.44 |
|   **Other** | 0 (0) | 1 (12.5) | 0.44 |
| **Laboratory results** | | | |
|   **Abnormal C-reactive protein (CRP), n** | 2 (20) | 2 (25) | 1 |
|   **Abnormal Erythrocyte sedimentation rate, n** | 1 (10) | 0 (0) | 1 |
| **BACES Score[b]** | | | |
|   **0** | 1 (10) | 0 (0) | 1 |
|   **1** | 7 (70) | 4 (50) | 0.63 |
|   **2** | 2 (20) | 3 (37.5) | 0.61 |
|   **3** | 0 (0) | 0 (0) | 0 |
|   **4** | 0 (0) | 1 (12.5) | 0.44 |
| **Mycobacteria species** | | | |
|   *M. avium* | 7 (70) | 6 (75) | 1 |
|   *M. intracellulare* | 0 (0) | 0 (0) | 1 |
|   *M. abscessus* complex[c] | 1 (10) | 2 (25) | 0.56 |
|   *M. chelonae* | 1 (10) | 0 (0) | 1 |
|   *M. kansasii* | 1 (10) | 0 (0) | 1 |
| **Radiologic features** | | | |

*(Continued)*

**Table 2.** (Continued)

| | Subjects, no. (%) | | |
| --- | --- | --- | --- |
| | **Non progressive NTM-LD (n = 10)** | **Progressive NTM-LD (n = 8)** | **P Value[a]** |
| **Demographics** | | | |
| Nodular/bronchiectasis | 10 (100) | 8 (100) | 1 |
| Cavitary | 0 (0) | 4 (50) | 0.02 |
| **Culture progress** | | | |
| Sputum/culture conversion[d] | 5 (50) | 6 (75) | 0.60 |
| Culture persistence[e] | 2 (20) | 2 (25) | 1 |
| **NTM treatment status[f]** | | | |
| Previous treatment completion | 1 (10) | 1 (12.5) | 1 |
| Ongoing antibiotic treatment | 0 (0) | 3 (37.5) | 0.07 |
| Discontinued treatment | 0 (0) | 0 (0) | 1 |
| Antibiotic treatment naïve | 9 (90) | 4 (50) | 0.44 |

**Abbreviations:** COPD = chronic obstructive pulmonary disease; FEV1 = Forced expiratory volume in the first second, FVC = Forced vital capacity, DLCO = Diffusing capacity for carbon monoxide, TLC = Total lung capacity; IQR = Interquartile range.

Values are number (%) or median [IQR].

[a] P values for comparison between (Fisher's exact test or Mann-Whitney U when appropriate)

[b] BACES score [16]: body mass index, age, cavity, erythrocyte sedimentation rate, and sex

[c] Includes two patients with macrolide-susceptible *M. abscessus* complex (subsp. *massiliense*) in the progressive NTM lung disease group, and one patient with cystic fibrosis and macrolide-resistant *M. abscessus* complex (subsp. *abscessus)* lung disease without radiological and clinical progression but persistent sputum culture positive results.

[d] Culture conversion included four patients who converted respiratory cultures within 6 months and two patients who converted cultures within 12 months.

[e] Includes two patients with persistent sputum cultures for macrolide-susceptible MAC isolates, one patient with fibrocavitary and bronchiectatic lung disease who further deteriorate and passed away despite intensification of her antibiotic treatment, and another patient with fibronodular and bronchiectatic lung disease who achieved sputum culture conversion after 23 months following treatment intensification.

[f] Antibiotic treatment for NTM at the time of study blood sampling.

Candida, showed statistical significance differences between NTM-LD patients and control subjects (**Fig 2F-2I**). Therefore, these data suggest that MTB300-specific IFN-γ ELISpot has the best diagnostic accuracy in distinguishing NTM-LD from controls compared with other ELISpot methods.

## Patients with NTM-LD showed higher frequency of antigen stimulated CD4 and CD8 T-cells co-expressing CD25 and CD134 than control subjects

We measured the percentage of CD4 and CD8 T cells expressing co-stimulatory molecules CD25 and CD134 by FC in PBMC after *ex vivo* antigen challenge with RD1 peptides, PPD and MTB300 peptides in NTM-LD patients and control subjects. The general gating strategy is shown in **S1 Fig**. The percentages of CD4$^+$CD25$^+$CD134$^+$ T cells in a control subject, and non-progressive and progressive NTM-LD patients are shown in the upper right quadrants in each plot (**Fig 3**). The percentage of PPD-specific CD25$^+$CD134$^+$ in CD4 and CD8 T cells was significantly higher in NTM-LD patients compared with control subjects (**Fig 4B, 4G**). ROC analysis revealed that PPD-specific CD25$^+$CD134$^+$ in CD4 T cells had the highest specificity of 91.6% (95% CI, 61.5–99.7) and sensitivity of 66.6% (95% CI, 40.9–86.6) with an AUC of 0.7917 (**Fig 5B**, **Table 3**). Similarly, immunoprofiling of PPD-specific CD25$^+$CD134$^+$ in CD8 T cells showed a specificity of 83.3% (95% CI, 51.5–97.9) and sensitivity of 72.2% (95% CI, 46.5–90.3) to differentiate NTM-LD cases from controls (**Fig 5B**, **Table 3**). The positive control anti-CD3

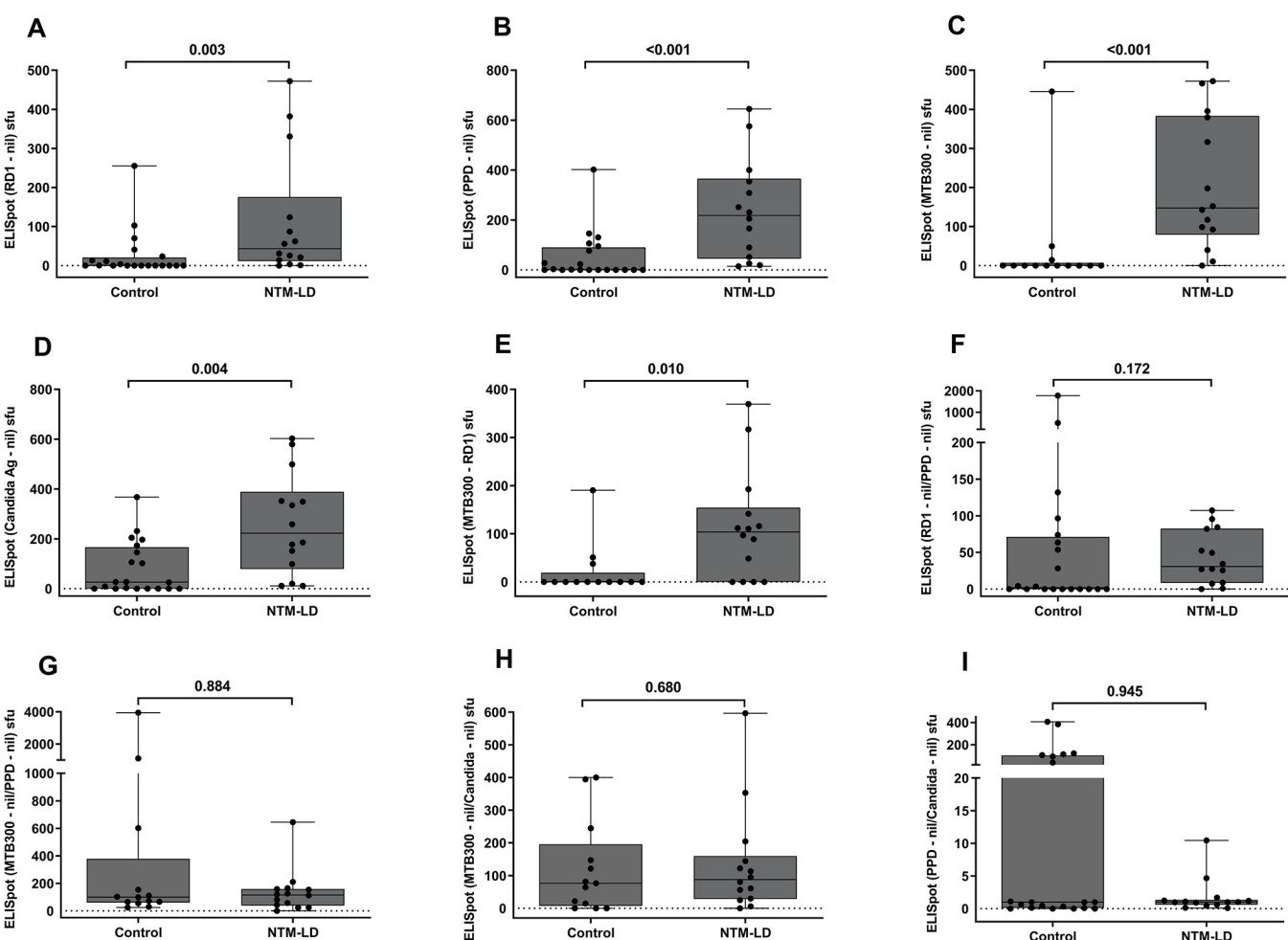

**Fig 2. Testing results of IFN-γ ELISpot in controls and NTM-LD patients.** IFN-γ ELISpot results of RD1 peptides (A), PPD (B), MTB300 peptide pool (C) and Candida antigen (D), MTB300-RD1 (E), ratios of net RD1/PPD (F), MTB300/PPD (G), MTB300/Candida (H) and PPD/Candida (I). The response by Ag-specific cells was background subtracted for each donor. Differences between the groups were compared using a Mann–Whitney U-test. Statistically significant differences were represented as p value. ns = nonsignificant (*P*>.05). The boxes show the median and interquartile range, and the whiskers show minimum and maximum values.

also yielded a specificity of 91.6% (95% CI, 61.5–99.7) but a low sensitivity of 50% (95% CI, 26–73.9) (**Fig 4E**). No significant differences were observed in the phenotypic frequencies with other antigen stimulations (**Fig 4A, 4C, 4D, 4F, 4H and 4I**).

## Distinguishing nonprogressive and progressive NTM-LD

We evaluated the value of IFN-γ ELISpot with our multiparametric antigen panel in the differentiation of progressor versus nonprogressor NTM-LD cases. A higher percentage of anti-CD3 stimulated IFN-γ producing cells was observed in nonprogressive NTM-LD when compared to progressive NTM-LD. While none of the ELISpot results with our four antigen stimulations attained a statistically significant difference between nonprogressive and progressive disease, there was a significantly higher IFN-γ response in progressive versus nonprogressive NTM-LD when we calculated the ratios of RD1-nil/PPD-nil and RD1-nil/anti-CD3-nil, (**Fig 5D and 6B**). ROC analysis showed that both RD1-nil/PPD-nil and RD1-nil/anti-CD3-nil provided the AUC of 0.854 with a sensitivity of 83.3% (95% CI, 35.8–99.5) and specificity of 87.5%

**Table 3. Diagnostic parameters of IFN-γ ELISpot and FC immunoprofiling across study groups.**

| Controls vs. NTM-LD infection | Phenotype | Sensitivity % | Specificity % | AUC | Cut-off |
|---|---|---|---|---|---|
| | ELISpot (RD1-nil) | 64.2 (CI 35.1–87.2) | 80 (CI 56.3–94.2) | 0.7982 | >25 |
| | ELISpot (PPD-nil) | 64.2 (CI 35.1–87.2) | 80 (CI 56.3–94.2) | 0.8536 | >100 |
| | ELISpot (MTB300-nil) | 78.5 (CI 49.2–95.3) | 92.3 (CI 63.9–99.8) | 0.8791 | >71 |
| | ELISpot (Candida-nil) | 64.2 (CI 35.1–87.2) | 80 (CI 56.3–94.2) | 0.7929 | >175 |
| | ELISpot (MTB300-RD1) | 71.4 (CI 41.9–91.6) | 84.6 (CI 54.5–98) | 0.7802 | >43 |
| | CD3+CD4+/CD25+CD134+ (PPD-nil) | 66.6 (CI 40.9–86.6) | 91.6 (CI 61.5–99.7) | 0.7917 | >0.056 |
| | CD3+CD4+/CD25+CD134+ (antiCD3-nil) | 50 (CI 26.0–73.9) | 91.6 (CI 61.5–99.7) | 0.7593 | >23.54 |
| | CD3+CD8+/CD25+CD134+ (PPD-nil) | 72.2 (CI 46.5–90.3) | 83.3 (CI 51.5–97.9) | 0.7130 | >0.056 |
| Nonprogressive vs. Progressive NTM-LD | CD3+CD8+/CD25+CD134+ (MTB300-nil) | 75 (CI 34.9–96.8) | 90 (CI 55.5–99.7) | 0.8313 | >0.106 |
| | ELISpot (RD1-nil)/(PPD-nil) | 83.3 (CI 35.8–99.5) | 87.5 (CI 47.3–99.6) | 0.8542 | >41.8 |
| | ELISpot (MTB300-RD1) | 66.6 (CI 22.2–95.6) | 100 (CI 63.0–100) | 0.8125 | <24.18 |
| | ELISpot (MTB300-nil)/(Candida-nil) | 66.6 (CI 22.2–95.6) | 87.5 (CI 47.3–99.6) | 0.8333 | <58.46 |
| | ELISpot (RD1-nil)/(antiCD3-nil) | 83.3 (CI 35.8–99.5) | 87.5 (CI 47.3–99.6) | 0.8542 | >7.25 |

AUC Area under the ROC curve. 95% CI: 95% confidence interval.

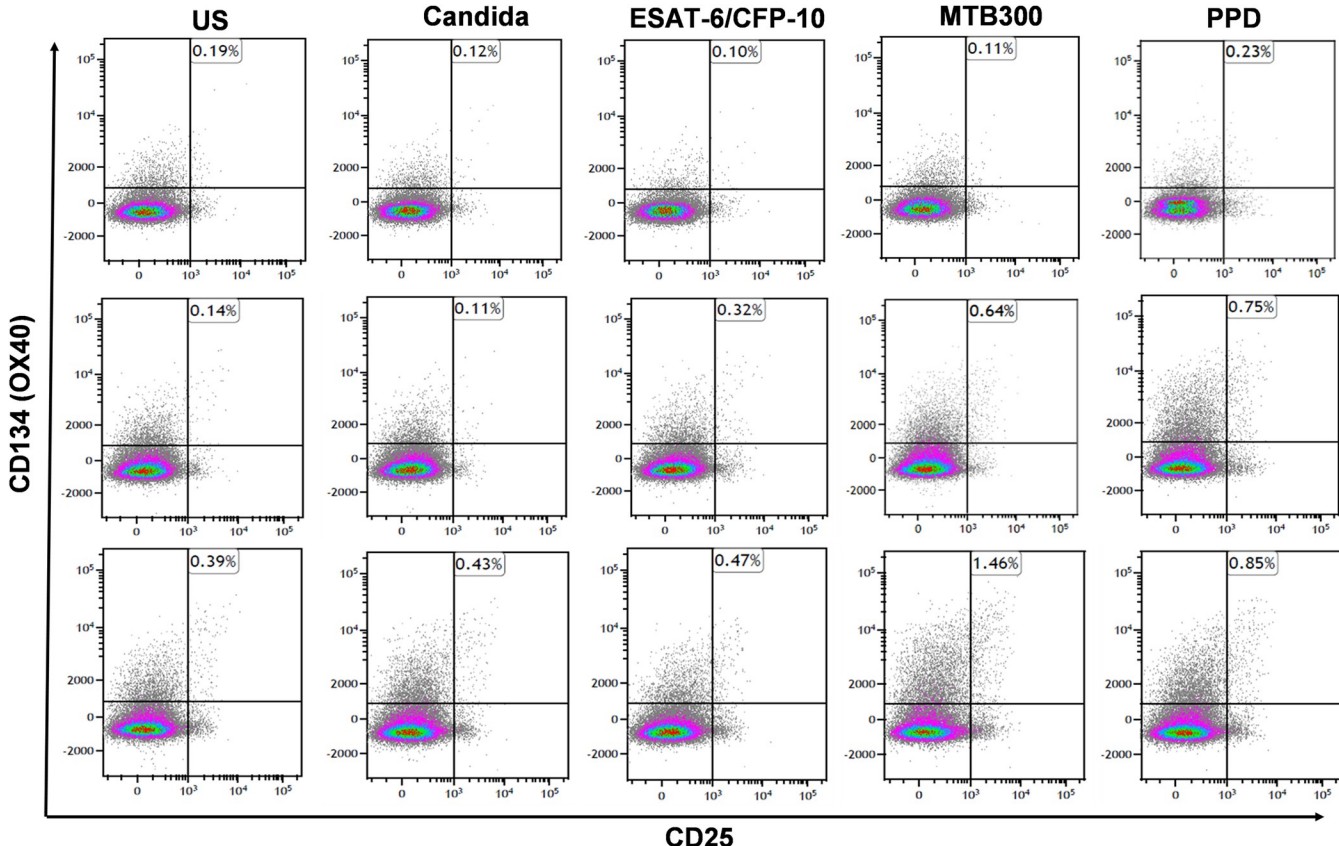

**Fig 3. Frequency of CD3+CD4+CD25+CD134+ T cells in controls, nonprogressive and progressive NTM-LD.** Representative flow cytometry plots showing the expression of CD3+CD4+CD25+CD134+ T cells in response to mycobacterial antigens in Controls, nonprogressive and progressive NTM-LD patients respectively. PBMCs of control subjects and NTM-LD patients were stimulated either with Candida antigen, ESAT-6/CFP-10 (RD1) peptides, MTB300 and PPD or left unstimulated for 40 hours and measured CD25 and CD134 response by flow cytometry. The percentage of CD4+CD25+CD134+ T cells was shown in the upper right quadrants in each plot. Upper panel–control subject (ID178); middle panel–nonprogressive NTM-LD (ID225); Lower panel–progressive NTM-LD (ID292).

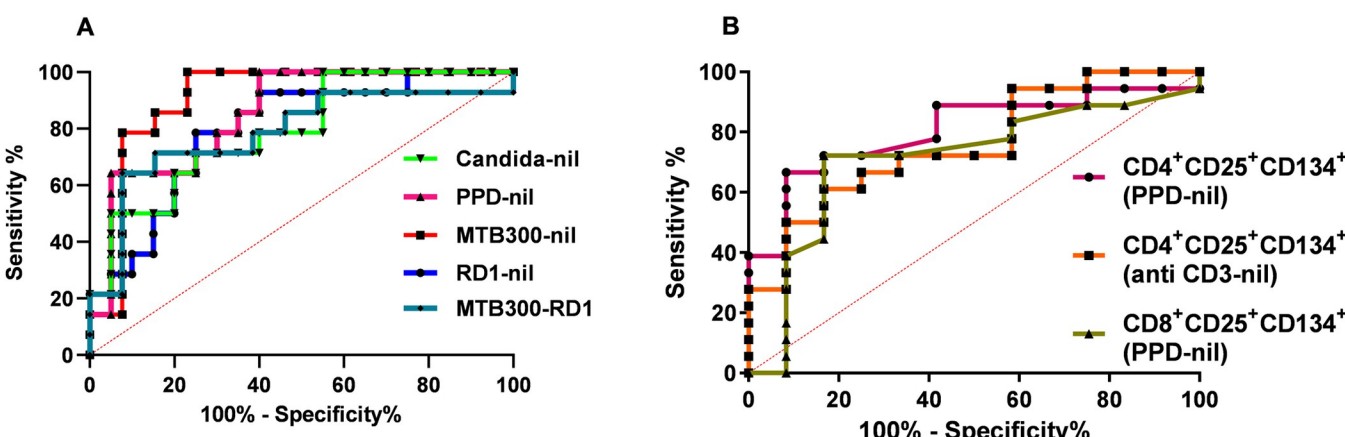

**Fig 4. Testing results of flow cytometric CD4+/CD8+CD25+CD134+ T cells in controls and NTM-LD cohorts.** Flow cytometric detection of the percentage of CD3+CD4+CD25+CD134+ against RD1 peptides (A), PPD (B), MTB300 peptide pool (C), Candida antigen (D) and anti-CD3 (E). Percentage of CD3+CD8+CD25+134+ in RD1 peptides (F), PPD (G), MTB300 peptide pool (H) and anti-CD3 (I). The response by stimulated cells was background subtracted for each donor. Differences between the groups were compared using a Mann–Whitney U-test. ns = nonsignificant ($P < 0.05$). Horizontal line represents median, and upper and lower boundaries of box represent 75th and 25th percentile. The whiskers extend from each quartile to the minimum and maximum.

**Fig 5. Diagnostic performance of IFN-γ ELISpot and CD25+CD134+ T cells in NTM-LD diagnosis.** Receiver operating characteristics curve (ROC) plots show the diagnostic accuracy of IFN-γ ELISpot (5A) and CD25+CD134+ markers (5B) in discriminating between control subjects and NTM-LD patients.

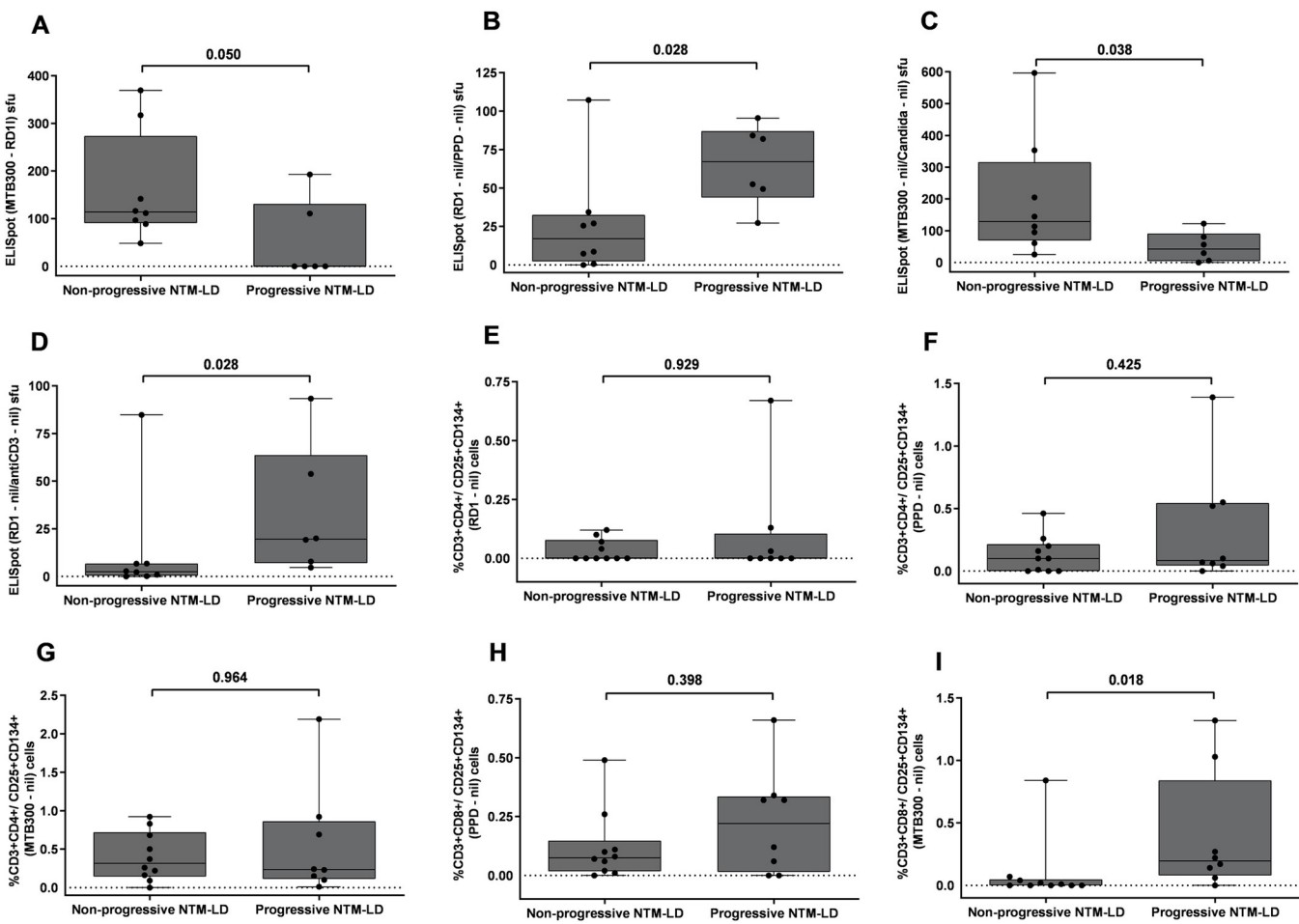

**Fig 6. Testing results of IFN-γ ELISpot and flow cytometric CD4⁺/CD8⁺CD25⁺CD134⁺ T cells in nonprogressive and progressive NTM-LD.** ELISpot results of MTB300-RD1 sfu (A), net ratios of RD1/PPD sfu (B), MTB300/Candida sfu (C) and RD1/antiCD3 sfu (D). Flow cytometric detection of percentage of CD3⁺CD4⁺CD25⁺CD134⁺ against RD1 peptides (E), PPD (F), MTB300 peptide pool (G). Percentage of CD3⁺CD8⁺CD25⁺134⁺ with PPD (H) and MTB300 peptide pool (I) The response by stimulated cells was background subtracted for each donor. Differences between the groups were compared using a Mann–Whitney U-test. ns = nonsignificant (*P*<0.05). The boxes show the median and interquartile range, and the whiskers show minimum and maximum values.

(95% CI, 47.3–99.6) to discriminate nonprogressive versus progressive NTM-LD (**Fig 7**, **Table 3**). In contrast, the ELISpot IFN-γ sfu values of MTB300-RD1 and the ratio of MTB300/Candida showed a significantly higher response in nonprogressive versus progressive NTM-LD (**Fig 6A–6C**). ROC curve analysis revealed that MTB300-RD1 had a diagnostic accuracy with an AUC of 0.813, a sensitivity of 66.6% (95% CI, 22.2–95.6) and a specificity of 100% (95% CI, 63–100) (**Fig 7**, **Table 3**). The sensitivity and specificity of MTB300/Candida ratio were 66.6% (95% CI, 22.2–95.6) and 87.5% (95% CI, 47.3–99.6), respectively (**Fig 7**, **Table 3**). Taken together, these data indicate that calculating the ELISpot ratios might be useful to distinguish progressive versus nonprogressive NTM-LD.

We also conducted a subgroup analysis by comparing patients with nonprogressive and progressive MAC lung disease since most of the NTM-LD patients were infected by macrolide-susceptible MAC isolates (13 out of 18 [72.2%] patients). We observed upward trends with two ELISpot ratios (RD1/PPD, RD1/antiCD3) and a FC (CD8⁺CD25⁺CD134⁺ [MTB300-nil] T-cells) assay in progressors vs. nonprogressors MAC lung disease, but P values did not attain level of significance by comparison across these subgroups (**S2 Fig**).

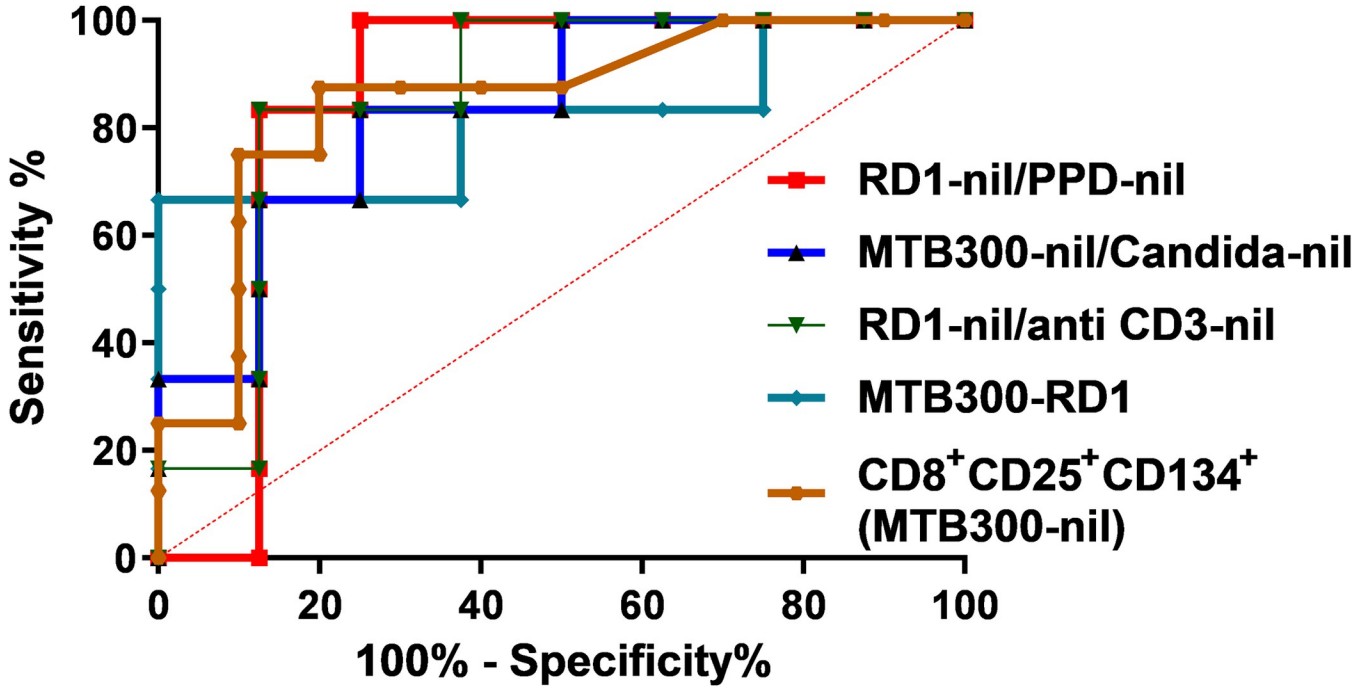

**Fig 7. Diagnostic performance of IFN-γ ELISpot and CD8⁺CD25⁺CD134⁺ for predicting progressive NTM-LD.** Receiver operating characteristics curve (ROC) were performed for ratios of IFN-γ ELISpot RD1/PPD, MTB300/Candida, RD1/anti-CD3 sfu, and MTB300-RD1 sfu; and MTB300 specific FC CD8⁺CD25⁺CD134⁺ markers in distinguishing progressive from nonprogressive NTM-LD patients.

Interestingly, the median percentage of CD4⁺CD25⁺CD134⁺ were slightly higher in nonprogressive than progressive NTM-LD after *ex vivo* stimulations with RD1, PPD and MTB300 antigens (**Fig 6E–6G**). In contrast, the percentage of CD8⁺ T-cells expressing CD25⁺CD134⁺ was increased in progressive versus nonprogressive NTM-LD cases after to PPD and MTB300 antigen stimulations (**Fig 6H and 6I**). However, only MTB300-specific CD8⁺CD25⁺CD134⁺ T cells displayed a significant difference between nonprogressive and progressive NTM-LD with a sensitivity of 75% (95% CI: 34.9–96.8), specificity of 90% (95% CI: 55.5–99.7), and an AUC of 0.8313 (**Fig 7, Table 3**). None of the other antigen stimulations or T cell subsets showed significant differences between these two groups.

## Discussion

The diagnosis of NTM-LD can be delayed or missed as some patients exhibit non-specific symptoms for several months or years before the diagnosis is made, and for some patients, diagnosis comes after the disease has progressed substantially [17, 18]. Our study findings suggest that a blood-based multiparametric IFN-γ ELISpot and flow cytometric immunoprofiling approach with mycobacterial antigens were not only able to accurately identify patients with NTM-LD, but also able to accurately differentiate patients with progressive from nonprogressive NTM-LD in whom antibiotics were deferred. These study findings are important for treatment management of patients with NTM-LD which remains challenging, often requiring prolonged treatment with three or more antibiotics, and frequently resulting in intolerance to these treatments [7, 19]. In addition, about 50% of patients with MAC lung disease achieve culture conversion without antibiotics, and others remain clinically and radiologically stable despite the presence of mycobacteria in sputum [20]. There is currently no reliable way to *a priori* differentiate which patients will improve with or without antibiotics. Some of these

NTM-LD patients do not produce adequate sputum quality for testing. Therefore, there is an urgent need to develop new non-sputum biomarkers and accurate testing methods not only to improve diagnosis but also to predict disease progression to inform individualized antimicrobial management for the patients who would benefit the most.

Despite recent progress made in developing risk scores to predict mortality associated with NTM-PD, little progress has been made to predict individual disease progression in patients without cavitary and/or severe disease. The BACES score is composed of low BMI (<18.5 kg/m²), advanced age (≥65 years), presence of cavity, elevated ESR, and male sex which can predict mortality in patients with NTM pulmonary disease [16, 21]. However, most of these were not significantly associated with progressive disease in our study, and may not directly affect the NTM-LD progression. It has been reported that the impact of BMI was inversely related to NTM-PD development [22, 23]. Consistent with these reports, we also observed that low BMI was significantly associated with progressive NTM-LD (p-0.05). Further, cavitary lesions were significantly associated with progressive NTM-LD (p-0.02) as 4 out of 8 patients showed progressive cavitary lesions that were also associated with low BMI which is consistent with a previous study [23]. Sputum culture results and serological markers have limited potential in identifying progressive NTM-LD disease that requires prompt initiation of antibiotics, which is commonly based on poorly defined patient's clinical, radiological and microbiological characteristics.

The current knowledge of the risk of disease progression and predictors remains limited. Recently, a study reported that high serum levels of Krebs von den Lungen-6 (KL-6) was associated with disease progression in MAC lung disease [24]. Further, systemic inflammatory markers, history of TB, cough, weight loss, presence of cavity and malnutrition were also reported as predictive for disease deterioration in NTM-LD [21, 22, 25, 26]. However, the results are highly heterologous and depend on the study patients' characteristics, the study settings, and lack of controlled clinical trials renders them inconclusive. Further, the number of studies on immune-based biomarkers, particularly antigen-specific T cell assays, for predicting disease progression are sparce [27, 28]. Here, our pilot study shows that this new strategy using a multiparametric IFN-γ ELISpot and FC immunoprofiling with a panel of mycobacterial antigens has the potential to improve the diagnosis of NTM infections and *a priori* infer risk of progressive versus nonprogressive NTM-LD, which can potentially assist individualized treatment management.

In contrast with other immunoprofiling studies, we have used both nonspecific mycobacterial antigens of purified protein derivatives (PPD) and more specific TB antigens such as the RD1-derived peptide pool and the MTB300 peptide megapool. Since 80% of proteins with significant homology are shared between PPDs of different species of NTM and *Mycobacterium tuberculosis* (*M. tb*), cross-reactive immunoassay responses are not unexpected [29, 30]. In contrast to crude *M. tb* lysate, or culture filtrate proteins, MTB300 contains equimolar concentrations of peptides which are efficiently processed and presented by antigen-presenting cells. This MTB300 peptide megapool contains a mixture of 300 *M. tb* derived T cell epitopes from 90 *M. tb* proteins that specifically targets a large fraction of *M. tb*-specific CD4$^+$ and probably CD8$^+$ T cells, which can share epitopes with NTM species [31, 32]. In our study, the overall response against PPD and MTB300 was higher in patients with progressive versus nonprogressive NTM-LD, possibly representing a higher antigen encounter by CD8$^+$ T-cells in peripheral blood associated with more progressive forms of NTM-LD.

Several previous studies examined the Th1, Th2 and Th17 cellular immune response responses by ELISA and FC in individuals with NTM-LD, although the methodologies employed have varied widely and the findings have been inconsistent [33]. Further, a commercially available anti-glycopeptidolipid (GPL)-core immunoglobulin A (IgA) antibody

measurement is approved overseas as a diagnostic tool for pulmonary MAC disease [34–36]. However, little is known about the clinical utility of the test for assessing disease progression in western countries. Interestingly, our multiparametric IFN-γ ELISpot assay with *ex vivo* stimulations with RD1, PPD, MTB300, and Candida antigens showed no significant difference between control subjects and NTM-LD patient groups and these results align with prior reports [37, 38]. A recent large retrospective study from China showed that ELISpot was effective in discriminating NTM from pulmonary TB but not between NTM-LD and controls [37]. Because of the opposite trends of IFN-γ response with different antigen conditions in NTM-LD and controls, we analyzed the ratio of various combinations of antigen stimulations. Calculation of ratios can eliminate the impact of individual IFN-γ variations on the ELISpot assay, and is also less affected by underlying host immune status [39, 40]. In our multiparametric IFN-γ ELISpot assay we found that ratios of RD1/PPD, RD1/anti-CD3, and MTB300/ Candida can significantly differentiate nonprogressive and progressive NTM-LD. Therefore, the calculation of antigen ratios seems to be useful to predict disease progression rather than measuring individual antigen specific IFN-γ responses. Overall, our findings suggest that the ratios of immune response to mycobacterial-specific and non-specific antigen stimulations may show other important aspects of host immune response in patients with progressive disease.

In addition, the CD25/CD134 assay has shown greater sensitivity in detecting *M. tb*-specific immune responses in late-stage HIV compared to QFT-GIT, which suggests that this non-cytokine immunoprofiling is relatively less affected by immunosuppression [41, 42]. It has been demonstrated that the upregulation of CD25/CD134 after RD1 antigen stimulation occurs in active TB patients with or without HIV infection [41]. We previously reported that a non-cytokine activation immunoassay strategy was able to differentiate latent TB patients from TB-uninfected controls and potentially infer risk of TB reactivation by predictive modeling [12]. Recently, Lindestam Arlehamn CS *et al.*, also reported that MAC-PD patients showed significantly increased MTB300-specific CD134+PD-L1+ co-expression in T-cells compared to QFT-positive controls [32]. In line with these studies, patients with progressive NTM-LD had a higher frequency of CD8 T-cells co-expressing CD25+CD134+ compared with patients with nonprogressive NTM-LD. Thus, MTB300 specific CD25+CD134+ expression is an independent factor of disease progression that helps to identify patients requiring treatment initiation. Taken together, these findings suggest that CD25+CD134+ expression on CD8+ T cells may represent a surrogate peripheral blood marker for unsuccessful control of NTM-LD.

Our study has some limitations. The first being the small number of NTM-LD patients and control subjects, which is explained by the pilot design of this work. Secondly, among 18 NTM-LD patients, most of them had MAC lung disease, and a subgroup analysis of this smaller group of patients did not achieve significance difference between progressive and nonprogressive MAC lung disease. However, two ELISpot ratios (RD1/PPD, RD1/antiCD3) and a FC (CD8+CD25+CD134+ [MTB300-nil] T-cells) assay showed an upward trend in progressive MAC lung disease. This subgroup analysis can lead to type II statistical error due to a small sample size. Therefore, our results should be further tested in a larger cohort that includes not only a larger number of patients with MAC lung disease but also other non-MAC NTM species. Third, we were unable to select age-matched controls for each case. The mean age of the patients with NTM-LD was higher than that of the controls, possibly leading to some bias, nevertheless, we did not observe substantial differences in the IFN-γ ELISpot and FC phenotypes with regard to age. Fourth, in the absence of a gold standard test for NTM exposure, we could not completely rule out prior NTM exposures or NTM infection in the control subjects; however, all of them had negative chest X-rays and no clinical symptoms or radiological signs of NTM-LD. Fifth, we examined and compared immunoprofiling data in mostly asymptomatic

control individuals but additional testing and validation research work would need to include patients with NTM-LD mimickers such as patients with non-NTM pneumonia and fungal lung infections, as well as patients with non-infectious lung conditions such as pulmonary malignancies, sarcoidosis and other inflammatory lung diseases. Finally, although this study was conducted in a specialized clinic that shares the same referral practice, the physicians' recommendations and patients' preferences may vary, which can influence treatment decisions and timing to initiate antimicrobials for NTM-LD, and thus, influence some of the progressive versus nonprogressive study designations in this retrospective study.

## Conclusions

Our pilot study results suggest that PPD-specific CD25$^+$CD134$^+$ upregulation in T-cells is a potential blood biomarker to accurately diagnose NTM-LD. Further, the percentage of MTB300-specific CD8$^+$CD25$^+$CD134$^+$ in CD8$^+$ T-cells and a multiparametric IFN-γ ELISpot assay can also accurately differentiate disease progression in NTM-LD. To our knowledge, this is the first study to use this immunoprofiling approach to differentiate patients with progressive and nonprogressive NTM-LD. Therefore, our findings suggest that our multiparameter diagnostic strategy with FC and IFN-γ ELISpot assays can not only provide a non-sputum-based diagnostic method to accurately identify NTM-LD patients, but also differentiate progressive disease status from nonprogressive NTM-LD. This novel diagnostic strategy could assist with optimal treatment management in NTM-LD. However, prospective and larger studies would be needed to test and validate this novel immunoprofiling approach.

## Supporting information

**S1 Fig. Representative gating strategy of CD4 and CD8 T cells.** Representative gating strategy followed for gating CD3 from lymphocyte population. CD3 T cells gated in the live singlet gate of PBMC and subsequently CD4 and CD8 T cells were gated.
(TIF)

**S2 Fig.** Testing results of IFN-γ ELISpot and flow cytometric CD4$^+$/CD8$^+$CD25$^+$CD134$^+$ T cells in nonprogressive and progressive MAC-LD: ELISpot results of MTB300-RD1 sfu (A), net ratios of RD1/PPD sfu (B), MTB300/Candida sfu (C) and RD1/antiCD3 sfu (D). Flow cytometric detection of percentage of CD3$^+$CD4$^+$CD25$^+$CD134$^+$ against RD1 peptides (E), PPD (F), MTB300 peptide pool (G). Percentage of CD3$^+$CD8$^+$CD25$^+$134$^+$ with PPD (H) and MTB300 peptide pool (I). The response by stimulated cells was background subtracted for each donor. Differences between the groups were compared using a Mann–Whitney U-test. The boxes show the median and interquartile range, and the whiskers show minimum and maximum values.
(TIF)

**S1 Data. Supplementary minimal dataset information.**
(XLSX)

## Acknowledgments

### Other contributions

We are grateful to the study participants and their families, the staff of the Mayo Clinic Institutional Review Board, the Pulmonary Research Unit and the staff and colleagues of the Mayo

Mycobacterial and Bronchiectasis Clinic, the Mayo Clinic Center for Tuberculosis, and the Mayo Infectious Disease Serology Laboratory for their diligent and valuable support.

## Author Contributions

**Conceptualization:** Kelly M. Pennington, Tobias Peikert, Patricio Escalante.

**Data curation:** Balaji Pathakumari, Thomas M. Cox, Virginia P. Van Keulen, Maleeha Shah, Mounika Vadiyala, Pedro Arias-Sanchez, Snigdha Karnakoti, Elitza S. Theel, Patricio Escalante.

**Formal analysis:** Paige K. Marty, Balaji Pathakumari, Virginia P. Van Keulen, Courtney L. Erskine, Cecilia S. Lindestam Arlehamn, Tobias Peikert, Patricio Escalante.

**Funding acquisition:** Kelly M. Pennington, Patricio Escalante.

**Investigation:** Paige K. Marty, Balaji Pathakumari, Thomas M. Cox, Virginia P. Van Keulen, Courtney L. Erskine, Patricio Escalante.

**Methodology:** Paige K. Marty, Balaji Pathakumari, Virginia P. Van Keulen, Courtney L. Erskine, Elitza S. Theel.

**Project administration:** Kelly M. Pennington, Patricio Escalante.

**Software:** Pedro Arias-Sanchez.

**Supervision:** Tobias Peikert, Patricio Escalante.

**Validation:** Paige K. Marty, Balaji Pathakumari, Virginia P. Van Keulen, Courtney L. Erskine, Patricio Escalante.

**Visualization:** Balaji Pathakumari, Patricio Escalante.

**Writing – original draft:** Paige K. Marty, Balaji Pathakumari, Patricio Escalante.

**Writing – review & editing:** Paige K. Marty, Balaji Pathakumari, Thomas M. Cox, Virginia P. Van Keulen, Courtney L. Erskine, Maleeha Shah, Mounika Vadiyala, Pedro Arias-Sanchez, Snigdha Karnakoti, Kelly M. Pennington, Elitza S. Theel, Cecilia S. Lindestam Arlehamn, Tobias Peikert, Patricio Escalante.

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
