## [Decision Letter · Decision Letter 0]

21 Dec 2023

PONE-D-23-27463Multiparameter immunoprofiling for the diagnosis and differentiation of progressive versus nonprogressive nontuberculous mycobacterial lung disease – a pilot studyPLOS ONE

Dear Dr. Escalante,

Thank you for submitting your manuscript to PLOS ONE. After careful consideration, we feel that it has merit but does not fully meet PLOS ONE’s publication criteria as it currently stands. Therefore, we invite you to submit a revised version of the manuscript that addresses the points raised during the review process.

We look forward to receiving your revised manuscript.

Kind regards,

Mao-Shui Wang

Academic Editor

PLOS ONE

Journal Requirements:

"This work was supported by the Lucile Nelson Career Development Award in Pulmonary Research. Part of this work was supported by the CHEST Foundation (K.M.P., P.E.), and the National Institute of Allergy and Infectious Diseases at the National Institutes of Health [AI141591 to P.E.]. Part of this project was also supported by Grant Number UL1 TR000135 from the National Center for Advancing Translational Sciences (NCATS)."

3. We note that you have a patent relating to material pertinent to this article. Please provide an amended statement of Competing Interests to declare this patent (with details including name and number), along with any other relevant declarations relating to employment, consultancy, patents, products in development or modified products etc. Please confirm that this does not alter your adherence to all PLOS ONE policies on sharing data and materials, as detailed online in our guide for authors http://journals.plos.org/plosone/s/competing-interests by including the following statement: ""This does not alter our adherence to  PLOS ONE policies on sharing data and materials.” If there are restrictions on sharing of data and/or materials, please state these. 

Please note that we cannot proceed with consideration of your article until this information has been declared.

"P.E. and T.P., and their institution have filed two patent applications related to immunodiagnostic laboratory methodologies for latent tuberculosis infection (Patent numbers: 9678071 and 10401360), which are not included in this manuscript. To date, there has been no income or royalties associated with those filed patent applications. PE participated in a short-term advisory scientific board for DiaSorin Molecular in 2020, which was outside the scope of the submitted manuscript, and honorarium was paid to Mayo Clinic. E.S.T serves as a consultant for Roche Diagnostics (Basel, Switzerland), Euroimmun US (Mountain Lakes, NJ, USA), and Seriummune Inc. (Goleta, CA, USA) on topics outside the scope of this manuscript. P.E., T.P., and E.S.T. have no other conflicts to declare. P.K.M., B.P., T.M.C., V.P.V, C.L.E., M.S., M.V., P.A.S., S.K., K.M.P., and C. S. L. A. have no conflicts to declare."

Please confirm that this does not alter your adherence to all PLOS ONE policies on sharing data and materials, by including the following statement: "This does not alter our adherence to  PLOS ONE policies on sharing data and materials.” (as detailed online in our guide for authors http://journals.plos.org/plosone/s/competing-interests).  If there are restrictions on sharing of data and/or materials, please state these. 

Please note that we cannot proceed with consideration of your article until this information has been declared. 

Reviewers' comments:

Reviewer's Responses to Questions

**Comments to the Author**

1. Is the manuscript technically sound, and do the data support the conclusions?

Reviewer #1: Partly

Reviewer #2: Yes

2. Has the statistical analysis been performed appropriately and rigorously? 

Reviewer #1: No

Reviewer #2: Yes

3. Have the authors made all data underlying the findings in their manuscript fully available?

Reviewer #1: Yes

Reviewer #2: Yes

4. Is the manuscript presented in an intelligible fashion and written in standard English?

Reviewer #1: Yes

Reviewer #2: Yes

5. Review Comments to the Author

Reviewer #1: Comments to author

This study investigated antigen-specific immunoprofiling utilizing flow cytometry (FC) of activation-induced markers (AIM) and IFN-γ enzyme-linked immune absorbent spot assay (ELISpot) accurately identify patients with NTM-LD, and differentiate those with progressive NTM-LD from non-progressive NTM-LD. The authors concluded that this immunoprofiling is able to identify patients with NTM-LD and distinguish patients with progressive NTM-LD from those with non-progressive NTM-LD. While this study provides valuable information, there are several points to be elucidated as follows.

Minor Comments

#1 Results, Table 2

The authors mentioned that non-progressive NTM-LD was defined as clinical and radiological stability over least 24 months in patients not treated with antibiotics for NTM-LD. However, the proportion of patients with culture persistence was comparable in both groups (non- progressive NTM-LD and progressive NTM-LD). It would be helpful if the authors could add details of the patients with progressive NTM-LD along with the reasons why the proportion of patients with ongoing antibiotic treatment or antibiotics treatment naïve was low in progressive NTM-LD group.

#2 Results, Tables, Figures

The authors included NTM-LD patients caused by different NTM species.

Each NTM species has different clinical features including radiological findings.

It would be needed to analyze and add data only on NTM patients caused by MAC (i.e. non-progressive MAC-LD vs. progressive MAC-LD).

Reviewer #2: The work presented here aims to evaluate antigen-specific immunoprofiling to identify patients with non-progressive or progressive non-tuberculous lung disease. The concept of the work is quite new and the clinical relevance of mycobacterioses fully supports such studies.

The flow of the paper is good, the manuscript is well-written and does not require professional proofreading before publication.

The title is informative, with a clear objective in line with the content of the article.

The summary tables, clearly and legibly presented, and all figures are a strong part of the article.

All references are correctly cited in the text.

My comments and questions are as follows:

1. There is no information on whether ELISpot or flow cytometry analyses were performed in duplicate or not.

2. Table 1 provides information on the aetiological factors identified in the patients studied. Did the authors compare the immunological profile of patients infected with different mycobacterial species?

3. What was the patient recruitment scheme? Please describe in more detail the inclusion and exclusion criteria of the study. A chart explaining these criteria would be desirable.

4) Why was another control group not included in the study, such as patients with non-mycobacterial pneumonia?

6. PLOS authors have the option to publish the peer review history of their article (what does this mean?). If published, this will include your full peer review and any attached files.

Reviewer #1: No

Reviewer #2: No

---

## [Author Response · Author response to Decision Letter 0]

14 Feb 2024

Response to Editor and reviewers’ comments

Editor Requirements:

1. Please ensure that your manuscript meets PLOS ONE's style requirements, including those for file naming

Response: We followed the PLOS ONE style templates as instructed, including the Figures and file naming.

2. Financial disclosure edits request: 

Response: This work was supported by the Lucile Nelson Career Development Award in Pulmonary Research. Part of this work was supported by the CHEST Foundation (K.M.P., P.E.), and the National Institute of Allergy and Infectious Diseases at the National Institutes of Health [AI141591 to P.E.]. Part of this project was also supported by Grant Number UL1 TR000135 from the National Center for Advancing Translational Sciences (NCATS). The funders had no role in study design, data collection and analysis, decision to publish, or preparation of the manuscript.

3. Request for amended statement of Competing Interests to declare about disclosed patent

Response: Please see amended text below (Editor’s request #4)

4. Competing Interests section edits request: 

Response: P.E. and T.P., and their institution have filed two patent applications related to immunodiagnostic laboratory methodologies for latent tuberculosis infection (Patent numbers: 9678071 and 10401360), which are not included in this manuscript. To date, there has been no income or royalties associated with those filed patent applications. This does not alter our adherence to PLOS ONE policies on sharing data and materials. PE participated in a short-term advisory scientific board for DiaSorin Molecular in 2020, which was outside the scope of the submitted manuscript, and honorarium was paid to Mayo Clinic. E.S.T serves as a consultant for Roche Diagnostics (Basel, Switzerland), Euroimmun US (Mountain Lakes, NJ, USA), and Seriummune Inc. (Goleta, CA, USA) on topics outside the scope of this manuscript. P.E., T.P., and E.S.T. have no other conflicts to declare. P.K.M., B.P., T.M.C., V.P.V, C.L.E., M.S., M.V., P.A.S., S.K., K.M.P., and C. S. L. A. have no conflicts to declare.

5. PLOS requires an ORCID iD for the corresponding author

Response: Corresponding Authors (Patricio Escalante) ORCID ID number: #000-002-4945-1035

We included the ORCID ID from the corresponding author in this revision manuscript version.

6. Please include captions for your Supporting Information files at the end of your manuscript, and update any in-text citations to match accordingly.

Response: We included captions for our manuscript supporting information.

Reviewers Comments to the Author

Reviewer #1: Comments to author

“This study investigated antigen-specific immunoprofiling utilizing flow cytometry (FC) of activation-induced markers (AIM) and IFN-γ enzyme-linked immune absorbent spot assay (ELISpot) accurately identify patients with NTM-LD, and differentiate those with progressive NTM-LD from nonprogressive NTM-LD. The authors concluded that this immunoprofiling is able to identify patients with NTM-LD and distinguish patients with progressive NTM-LD from those with nonprogressive NTM-LD. While this study provides valuable information, there are several points to be elucidated as follows.”

Minor Comments

#1 Results, Table 2

The authors mentioned that nonprogressive NTM-LD was defined as clinical and radiological stability over least 24 months in patients not treated with antibiotics for NTM-LD. However, the proportion of patients with culture persistence was comparable in both groups (non- progressive NTM-LD and progressive NTM-LD). It would be helpful if the authors could add details of the patients with progressive NTM-LD along with the reasons why the proportion of patients with ongoing antibiotic treatment or antibiotics treatment naïve was low in progressive NTM-LD group.

Response: We appreciated the reviewer’s comments and carefully reviewed our study data and new available outside electronic medical records to complement, verify, and further update Table 2. Only 2 out of 8 patients (25%) with progressive NTM-LD had persistent positive sputum cultures. Although this is a small subgroup analysis, this low proportion is within expected rates for treatment outcomes in patients with MAC lung disease, which most of these patients with progressive NTM-LD were in our pilot study (6 out of 8 patients). Treatment success rates defined by sustained sputum culture conversion varies between 60 to 83% in most studies (Kwak N et al. Clin Infect Dis 2017; Jeong BH, et al. Am J Respir Crit Care Med 2015; Wallace RJ Jr, et al. Chest 2014) which can be explained by several factors, including host, disease phenotypes and treatment characteristics (Loebinger MR. Eur Respir J 2017). We also included two patients with M. abscessus complex subsp. massiliense in this progressive NTM group, which also usually have satisfactory treatment outcomes to guidelines-based antibiotic regimens that includes macrolides (Pasipanodya JG et al. Antimicrob Agents Chemother 2017). 

In addition, we noticed two errors in Table 2 that were corrected, and we apologize for these mistakes. One error was in the sputum conversion rate for the progressive NTM group. 6 out of 8 patients (but not 2 out of 8 as originally written) achieved culture conversion at 6 to 12 months after initiation of antimicrobial therapy. Also, one patient in this group had previous treatment completion and we adjusted the P values for these updated baseline study groups analysis in Table 2. We also added more detailed clinical and microbiological information for these patients with progressive NTM-LD in Table 2 as requested by the reviewer.

#2 Results, Tables, Figures

The authors included NTM-LD patients caused by different NTM species.

Each NTM species has different clinical features including radiological findings.

It would be needed to analyze and add data only on NTM patients caused by MAC (i.e. nonprogressive MAC-LD vs. progressive MAC-LD).

Response: We agree with Reviewer 1 that there is heterogeneity in terms of clinical presentations and radiological findings among different NTM species. However, our pilot study included mostly patients with MAC lung disease (72%) and only a few patients with lung disease associated with other pathogenic NTM species but with similar clinical characteristics as described in more detail in the updated Table 2. Of note, even among patients with the same NTM species there is heterogeneity in clinical presentations, radiological findings, and disease phenotypes. However, we carefully grouped this study cohort in patients within progressive and nonprogressive NTM lung disease to address our main biomarker study question since these 2 disease trajectories can be shared across patients with other common pathogenic NTM species. A similar study design approach was taken to study radiographic progression for NTM lung disease across various pathogenic NTM species diagnosed in patients by bronchoscopy in Taiwan (Huang HL, et al., Respir Med 2020). 

For completeness, we decided to conduct a subgroup analysis by comparison of only patients with nonprogressive and progressive MAC lung disease as requested by Reviewer 1. We observed upward trends with two ELISpot ratios (RD1/PPD, RD1/antiCD3) and one FC (CD8+CD25+CD134+ [MTB300-nil] T-cells) assay results in progressors vs. nonprogressors MAC lung disease, but P values did not attain statistical level of significance probably due to type II error in this small subgroup comparison. We added a supplemental Figure (S2 Fig) and briefly described this subgroup analysis in the results section (Lines 289-294) and commented in the discussion section (Lines 413-419).

Reviewer #2: The work presented here aims to evaluate antigen-specific immunoprofiling to identify patients with nonprogressive or progressive non-tuberculous lung disease. The concept of the work is quite new and the clinical relevance of mycobacterioses fully supports such studies.

The flow of the paper is good, the manuscript is well-written and does not require professional proofreading before publication.

The title is informative, with a clear objective in line with the content of the article.

The summary tables, clearly and legibly presented, and all figures are a strong part of the article.

All references are correctly cited in the text.

Response: We appreciated all the supportive comments and feedback from Reviewer 2.

My comments and questions are as follows:

1. There is no information on whether ELISpot or flow cytometry analyses were performed in duplicate or not.

Response: Thank you for your important clarification question. We conducted the ELISpot assay in triplicate manner for each antigenic condition. Results were expressed as mean spot forming units (SFU)/well ± standard deviation (SD) of triplicates. Blank and negative control wells were included to ensure the quality of the assay and added test to the methods section to describe (Lines 113-118).

Considering the cell count, size of tested antigens, reagents cost and merely running small pilot study, we performed the flow cytometry in single well manner for each antigenic condition. However, we prepared the homogenous suspension of PBMC and maintained consistent cell density across the stimulations with appropriate controls to provide a reference point for data analysis with greater confidence and accuracy. We also described this in the methods section (Lines 149-151).

2. Table 1 provides information on the aetiological factors identified in the patients studied. Did the authors compare the immunological profile of patients infected with different mycobacterial species?

Response: This pilot study had very few patients with lung disease associated with NTM species other than MAC isolates (please see prior response to Reviewer 1), and it would be probably challenging to accurately determine statistical differences in patients with different NTM species.

3. What was the patient recruitment scheme? Please describe in more detail the inclusion and exclusion criteria of the study. A chart explaining these criteria would be desirable.

Response: This NTM immunoprofiling study is part of a larger study that characterize antigen-specific immunoassays and biomarkers to differentiate individuals with latent tuberculosis infection, which included study patients with NTM lung diseases as control subjects as we mentioned in our original cover letter. We added text in the methods section to further clarify (Lines 88-90). We also added text in the methods section (Lines 103-105) and a Flowchart (Fig 1) in results (Lines 167-168) to further describe the inclusion and exclusion criteria utilized in this NTM pilot study.

4) Why was another control group not included in the study, such as patients with non-mycobacterial pneumonia?

Response: This is another excellent clarification question from Reviewer 2. Our central hypothesis is that NTM lung disease can be accurately diagnose and disease trajectory can be differentiated by blood-based antigen-specific immune profiling. As the first step to test this hypothesis, we aimed to design and conduct a pilot study with samples from well-characterized NTM-LD patients and mostly asymptomatic QuantiFERON-negative individuals to test whether a similar immunoprofiling approach that we previously utilized to study M. tuberculosis infections, can also accurately identify NTM lung disease, and to differentiate disease trajectory. This initial pilot study has provided promising results towards meeting this research goal, but additional work is needed to test and validate these biomarkers and immunoprofiling methods in larger study cohorts that would include not only patients with non-mycobacterial pneumonia but also patients other NTM lung disease mimickers such fungal lung diseases, and non-infectious lung diseases including pulmonary malignancies. We added additional related comments in the discussion section (lines 426 to 431). We are currently undergoing such research efforts, but more work would be needed to study other patient populations in other locations.

---

## [Decision Letter · Decision Letter 1]

20 Mar 2024

Multiparameter immunoprofiling for the diagnosis and differentiation of progressive versus nonprogressive nontuberculous mycobacterial lung disease – a pilot study

PONE-D-23-27463R1

Dear Dr. Escalante,

We’re pleased to inform you that your manuscript has been judged scientifically suitable for publication and will be formally accepted for publication once it meets all outstanding technical requirements.

Kind regards,

Mao-Shui Wang

Academic Editor

PLOS ONE

Additional Editor Comments (optional):

Reviewers' comments:

Reviewer's Responses to Questions

**Comments to the Author**

1. If the authors have adequately addressed your comments raised in a previous round of review and you feel that this manuscript is now acceptable for publication, you may indicate that here to bypass the “Comments to the Author” section, enter your conflict of interest statement in the “Confidential to Editor” section, and submit your "Accept" recommendation.

Reviewer #1: All comments have been addressed

2. Is the manuscript technically sound, and do the data support the conclusions?

Reviewer #1: Yes

3. Has the statistical analysis been performed appropriately and rigorously? 

Reviewer #1: Yes

4. Have the authors made all data underlying the findings in their manuscript fully available?

Reviewer #1: Yes

5. Is the manuscript presented in an intelligible fashion and written in standard English?

Reviewer #1: Yes

6. Review Comments to the Author

Reviewer #1: The authors responded to the reviewer's comments and revised the manuscript well.

I think this manuscript will be acceptable.

7. PLOS authors have the option to publish the peer review history of their article (what does this mean?). If published, this will include your full peer review and any attached files.

Reviewer #1: No

---

## [Editor Report · Acceptance letter]

8 Apr 2024

PONE-D-23-27463R1 

PLOS ONE

Dear Dr. Escalante, 

I'm pleased to inform you that your manuscript has been deemed suitable for publication in PLOS ONE. Congratulations! Your manuscript is now being handed over to our production team.

Kind regards, 

on behalf of

Dr. Mao-Shui Wang 

Academic Editor

PLOS ONE